

# CDMPred: a tool for predicting cancer driver missense mutations with high-quality passenger mutations

Lihua Wang[1,2], Haiyang Sun[3], Zhenyu Yue[4], Junfeng Xia[1] and Xiaoyan Li[1]

[1] Information Materials and Intelligent Sensing Laboratory of Anhui Province, Institutes of Physical Science and Information Technology, Anhui University, Hefei, Anhui, China
[2] School of Information Engineering, Huangshan University, Huangshan, Anhui, China
[3] State Key Laboratory of Medicinal Chemical Biology, NanKai University, Tianjin, Tianjin, China
[4] School of Information and Artificial Intelligence, Anhui Agricultural University, Hefei, Anhui, China

## ABSTRACT

Most computational methods for predicting driver mutations have been trained using positive samples, while negative samples are typically derived from statistical methods or putative samples. The representativeness of these negative samples in capturing the diversity of passenger mutations remains to be determined. To tackle these issues, we curated a balanced dataset comprising driver mutations sourced from the COSMIC database and high-quality passenger mutations obtained from the Cancer Passenger Mutation database. Subsequently, we encoded the distinctive features of these mutations. Utilizing feature correlation analysis, we developed a cancer driver missense mutation predictor called CDMPred employing feature selection through the ensemble learning technique XGBoost. The proposed CDMPred method, utilizing the top 10 features and XGBoost, achieved an area under the receiver operating characteristic curve (AUC) value of 0.83 and 0.80 on the training and independent test sets, respectively. Furthermore, CDMPred demonstrated superior performance compared to existing state-of-the-art methods for cancer-specific and general diseases, as measured by AUC and area under the precision-recall curve. Including high-quality passenger mutations in the training data proves advantageous for CDMPred's prediction performance. We anticipate that CDMPred will be a valuable tool for predicting cancer driver mutations, furthering our understanding of personalized therapy.

# INTRODUCTION

Cancer is a leading cause of death and suffering in humans worldwide, resulting in nearly 20 million new cases alongside 9.7 million deaths in 2022 (*Bray et al., 2024*). Researchers have confirmed that cancer is a multifaceted genetic disease caused by the accumulation of numerous mutations in the genome (*Wood et al., 2007*; *Tomasetti et al., 2015*; *Xi et al., 2020*). However, the tumorigenesis and development of most cancers are primarily driven by a small number of critical mutations (*Hanahan & Weinberg, 2011*; *Muiños et al., 2021*; *Ostroverkhova, Przytycka & Panchenko, 2023*), while the remaining mutations are

Corresponding author
Xiaoyan Li, lixiaoyan@ahu.edu.cn

considered neutral (passengers). Identifying driver mutations from passenger mutations holds significant importance, as drivers are commonly utilized as diagnostic and prognostic biomarkers and potential drug targets for cancer treatment (*Xi et al., 2020*; *Cheng et al., 2024*).

*Vogelstein et al. (2013)* observed that most protein-coding mutations in cancer genomes were missense changes. Consequently, our focus in this study is on cancer driver missense mutations. To date, numerous computational methods have been developed to predict driver missense mutations, such as boostDM (*Muiños et al., 2021*), Cancer-specific High-throughput Annotation of Somatic Mutations (CHASM) (*Carter et al., 2009*), Transformed Functional Impact score for Cancer (transFIC) (*Gonzalez-Perez, Deu-Pons & Lopez-Bigas, 2012*), Cancer Driver Annotation (CanDrA) (*Mao et al., 2013*), Functional Analysis through Hidden Markov Models (FATHMM) (*Shihab et al., 2013*), CScape-somatic (*Rogers, Gaunt & Campbell, 2020*), and CHASMplus (*Tokheim & Karchin, 2019*). Additionally, some methods are focused on identifying driver mutations at critical sites, such as protein allosteric sites (*Song et al., 2019*; *Song et al., 2023*; *Shen et al., 2017*). These methods typically utilize positive samples obtained from cancer-related databases, such as the Catalogue of Somatic Mutations in Cancer (COSMIC) database (*Carter et al., 2009*; *Gonzalez-Perez, Deu-Pons & Lopez-Bigas, 2012*; *Mao et al., 2013*; *Shihab et al., 2013*; *Rogers, Gaunt & Campbell, 2020*), while negative samples are commonly derived from statistical methods (*Muiños et al., 2021*; *Carter et al., 2009*) or putative samples (*Gonzalez-Perez, Deu-Pons & Lopez-Bigas, 2012*; *Mao et al., 2013*; *Shihab et al., 2013*; *Rogers, Gaunt & Campbell, 2020*; *Tokheim & Karchin, 2019*).

This study evaluated the potential for improved driver prediction by investigating high-quality passenger mutations. We then proposed a predictor, CDMPred, which incorporates high-quality passenger mutations and utilizes the eXtreme Gradient Boosting (XGBoost) algorithm. Initially, we conducted comparative analyses of the Cancer Passenger Mutations database (dbCPM), which comprises highly curated passenger mutations (*Yue, Zhao & Xia, 2020*). The results indicated that the dbCPM data aligns with other negative datasets regarding most classical features, while exhibiting specificity for cancer-related features (*Yue, Zhao & Xia, 2020*; *Wong et al., 2011*). Subsequently, we employed the high-quality passenger mutation data for model training and encoded 65 features. We used feature importance to identify the top 10 features from the 65 features mentioned above and evaluated the performance of various machine learning algorithms on the training set. Ultimately, we employed the optimal model (CDMPred) with an XGBoost classifier and the top 10 features. The results obtained from the training and independent test sets demonstrated that CDMPred exhibited superior performance compared to several state-of-the-art methods for both cancer-specific and general diseases, as assessed by two threshold-independent metrics: the area under the receiver operating characteristic curve (AUC) and the area under the precision–recall curve (AUPR).

## MATERIALS & METHODS

Figure 1 presents the flowchart of the CDMPred method. Portions of this text were previously published as part of a preprint (https://www.researchsquare.com/article/rs-
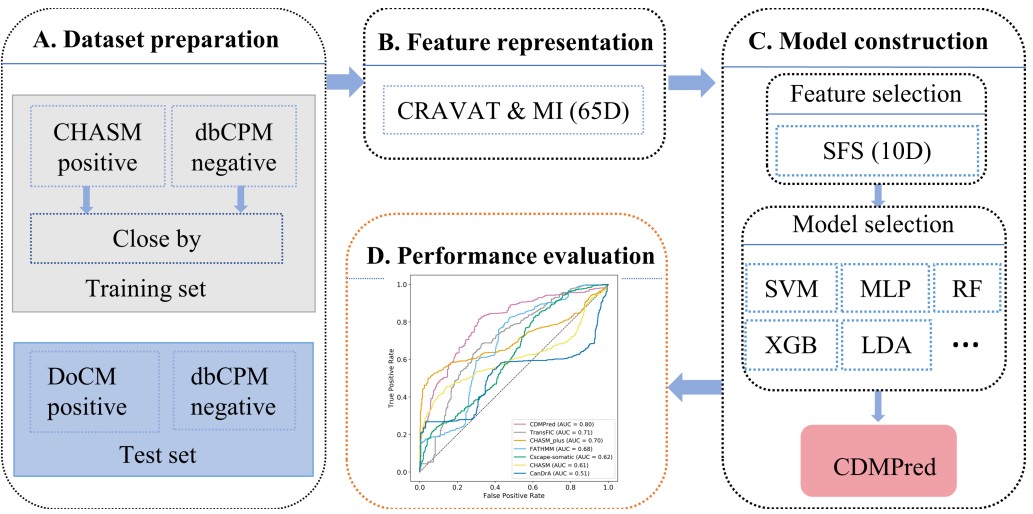

**Figure 1** **Flowchart of the proposed method.**

1350438/v1). The procedure consists of four steps: dataset preparation, feature representation, model construction, and performance evaluation. Each step is explained in detail below.

## Dataset preparation

The datasets were divided into two groups: one for feature analysis and the other for model construction and performance evaluation. One cancer driver mutation dataset and three passenger mutation datasets were used for feature analysis. For positive samples, we selected 1,248 driver missense mutations from the Database of Curated Mutations (DoCM) (v3.2) (*Ainscough et al., 2016*), which is a reliable source that aggregates functionally validated mutations in cancer. For negative samples, we gathered three passenger datasets. The dataset dbCPM (v1.1) consists of 1,919 passenger mutations, including 1,634 distinct missense mutations (*Yue, Zhao & Xia, 2020*). The other two datasets are oriented from classic prediction tools for cancer-specific driver mutations. Expressly, the dataset FATHMM was initially obtained from the UniProt database, which was taken as negative samples in the FATHMM training set (*Shihab et al., 2013*; *Apweiler et al., 2004*), and the dataset CHASM (v3.1) consists of synthetic passenger mutations in the CHASM training set (*Carter et al., 2009*). We removed the mutations simultaneously in DoCM in each passenger mutation dataset. The details are presented in Table S1.

The datasets utilized for model construction are described as follows. Out of the 1,634 missense passenger mutations in dbCPM v1.1, 1,104 items from dbCPM v1.0 were used as negative samples in our training set. We filtered the 13,235 positive samples in the CHASM (v3.1) training set to avoid overlap with samples from dbCPM v1.0. Next, we included only positive samples within 50 bp of a passenger mutation on the same transcript to address the imbalance and potential bias towards positive samples. As a result, our training set retained 2,151 driver missense mutations. We obtained an independent test

**Table 1  Summary of mutation datasets used for model construction and evaluation.**

|  | Training set | | Independent test set | |
| --- | --- | --- | --- | --- |
|  | **Positive** | **Negative** | **Positive** | **Negative** |
| Source | CHASM v3.1 | dbCPM v1.0 | DoCM v3.2 nonoverlap | dbCPM v1.1 nonoverlap |
| Number | 2151 | 1104 | 567 | 530 |

**Notes.**

DoCM v3.2 nonoverlap, data in DoCM v3.2 but not in CHASM v3.1; dbCPM v1.1 nonoverlap, data in dbCPM v1.1 except for dbCPM v1.0.

set to benchmark performance against state-of-the-art prediction tools. First, we collected missense mutations in dbCPM v1.1 reported after the initial database update (dbCPM v1.0) to serve as negative samples. Secondly, we considered all 1,248 driver mutations in DoCM as our positive samples. To prevent type 1 circularity (*Grimm et al., 2015*), which can cause overfitting from overlapping training and evaluation datasets, we excluded overlapping data with the training set, resulting in 567 driver mutations. The datasets utilized for model construction and performance evaluation are detailed in Table 1.

## Feature representation

Considering both the significance of the protein's functions and conservation, seven feature groups were provided to capture the specific characteristics of cancer driver mutations, comprising protein physicochemical properties, evolutionary conservation scores, exon features, protein local structures, regional composition, amino acid residue triplet features, and UniProt annotations. For each missense mutation in the datasets mentioned above, the features were encoded with the 85 pre-computed features available in SNVBox (*Wong et al., 2011*; *Won et al., 2021*) from a dockerized tool, the Cancer-Related Analysis of Variants Toolkit (*Masica et al., 2017*) (CRAVAT, version 5.2.3). To prepare the input data, we curated the transcript information using Ensembl GRCh37 (*Flicek et al., 2014*) as a reference. Each feature underwent scaling by subtracting the mean value and dividing it by the root mean square (RMS) value, utilizing pre-computed values for the entire genome. After CHASM (*Shen et al., 2017*; *Yue, Zhao & Xia, 2020*), we applied the information gain method to remove irrelevant features among the 85 candidate features. By using a uniform threshold, we selected 65 predictive features that possessed a minimum of 0.001 bits of mutual information Specifically, 13 out of 16 protein physicochemical properties, all six evolutionary conservation score features, all three exon features, 11 out of 12 protein local structure features, six out of 11 regional composition features, and 26 out of 28 UniProt annotations were included. The amino acid residue trimer features were also excluded. A detailed list is indicated in Table S2.

## Model construction

We utilized feature importance with XGBoost to select an optimal subset of features. Subsequently, we comprehensively evaluated multiple algorithms on the training set using a 10-fold cross-validation (*Buske et al., 2013*). We selected eight classifiers, namely random forest (RF), support vector machine (SVM), multilayer perceptron (MLP), gradient boosting decision tree (GBDT), linear discriminant analysis (LDA), logistic regression (LR),
naïve Bayes (NB), and XGBoost (*Chen & Guestrin, 2016*). All the algorithms mentioned above were implemented using scikit-learn (v0.22.2) and Python 3.7. The classifiers were implemented with parameters optimized through grid search, utilizing the 10-fold cross-validation results of the training set. Specifically, we optimized three critical parameters in XGBoost: the boosting learning rate (learning_rate), the maximum depth of the tree (max_depth), and the subsample ratio of columns when constructing each tree (colsample_bytree).

## Performance evaluation

As quantitative measurements of prediction results, we employed two threshold-independent measures: AUC and AUPR (*Carter et al., 2009*; *Dong et al., 2015*). Additionally, we used two qualitative measures, namely sensitivity (or true positive rate) and specificity (true negative rate), for model performance analysis, as previously described in research (*Cheng et al., 2020*; *Zeng et al., 2018*). These measures are defined as follows:

$$\text{Sensitivity} = \frac{\text{TP}}{\text{TP} + \text{FN}}$$

$$\text{Specificity} = \frac{\text{TN}}{\text{TN} + \text{FP}}$$

where TP (true positive) means the number of correctly predicted cancer driver mutations, FP (false positive) represents the number of passenger mutations predicted as drivers, TN (true negative) represents the number of correctly predicted passenger mutations, and FN (false negative) indicates the number of cancer driver mutations predicted as passengers.

The permutation test was conducted on CDMPred to demonstrate that the model learned more than noise. Specifically, we first trained the CDMPred model on the data and saved the AUC value of 10-fold cross-validation. Secondly, we randomly permuted the class labels in the dataset and trained a new model called "CDMPred_random". Thirdly, we assessed the performance of "CDMPred_random" regarding AUC. We repeated the second and the third steps 1,000 times. Finally, we calculated the empirical *p*-value by comparing the distribution of the 1,000 values to the corresponding value from the original CDMPred. The permutation test algorithm was implemented with the function named permutation_test_score in scikit-learn.

## RESULTS

### Analysis of features between different datasets

We quantified 85 features for all datasets presented in Table S1, which comprehensively represents the biological impacts of the mutation in the human genome (*Wong et al., 2011*). We statistically analyzed the dbCPM samples using these features in the nonparametric Wilcoxon signed rank hypothesis test. Figure 2A displays the significant features ($p < 0.05$) of positive samples obtained from DoCM, dbCPM, and other negative samples. Figure 2B illustrates the significant features among all negative samples. Our findings indicate that dbCPM data closely resemble other negative samples in terms of most classical features,

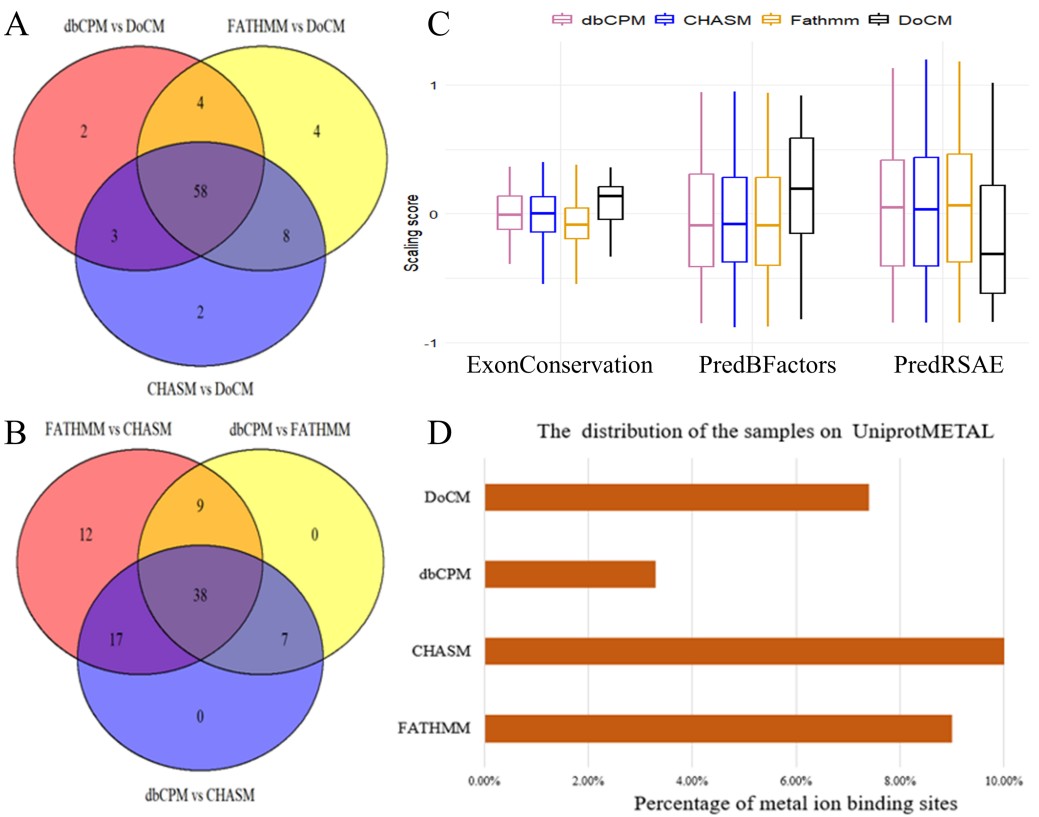

**Figure 2 Statistical analysis of samples.** (A) Overlap of features showing significant differences between negative and gold standard positive samples. (B) Overlap of features showing significant differences between negative samples. (C) RMS score distribution of all samples for three classical features. (D) RMS score distribution of all samples for the UniprotMETAL feature.

including 'ExonConservation' (conservation score for the entire exon calculated from the phylogenetic alignment of 46 species) and 'PredBFactorS' (probability that the residue backbone of wild type is stiff) (*Kent et al., 2002*; *Katzman et al., 2008*). Subsequently, we identified three features based on their *p*-values, and the RMS score distribution of all samples is presented in Fig. 2C. Therefore, the mutations in dbCPM were utilized as qualified negative samples for predicting disease-causing mutations. dbCPM exhibited distinguishable characteristics in cancer-specific features compared to other negative samples, including 'UniprotMETAL' (a binding site for a metal ion) and 'UniprotREP' (positions of repeated sequence motifs or domains) (*Ribeiro et al., 2004*; *Xu et al., 2016*). Figure 2D illustrates the distribution of RMS scores for the UniprotMETAL feature across all samples. These findings further support that dbCPM mutations are more representative than other negative samples in modeling a wide range of passenger mutations and are better suited for predicting cancer driver mutations.

## Explorations for an optimal model

Figure 3 displays the AUC values of the training set for the eight classifiers. XGBoost outperformed all other classifiers, achieving an AUC value of 0.82. XGBoost was

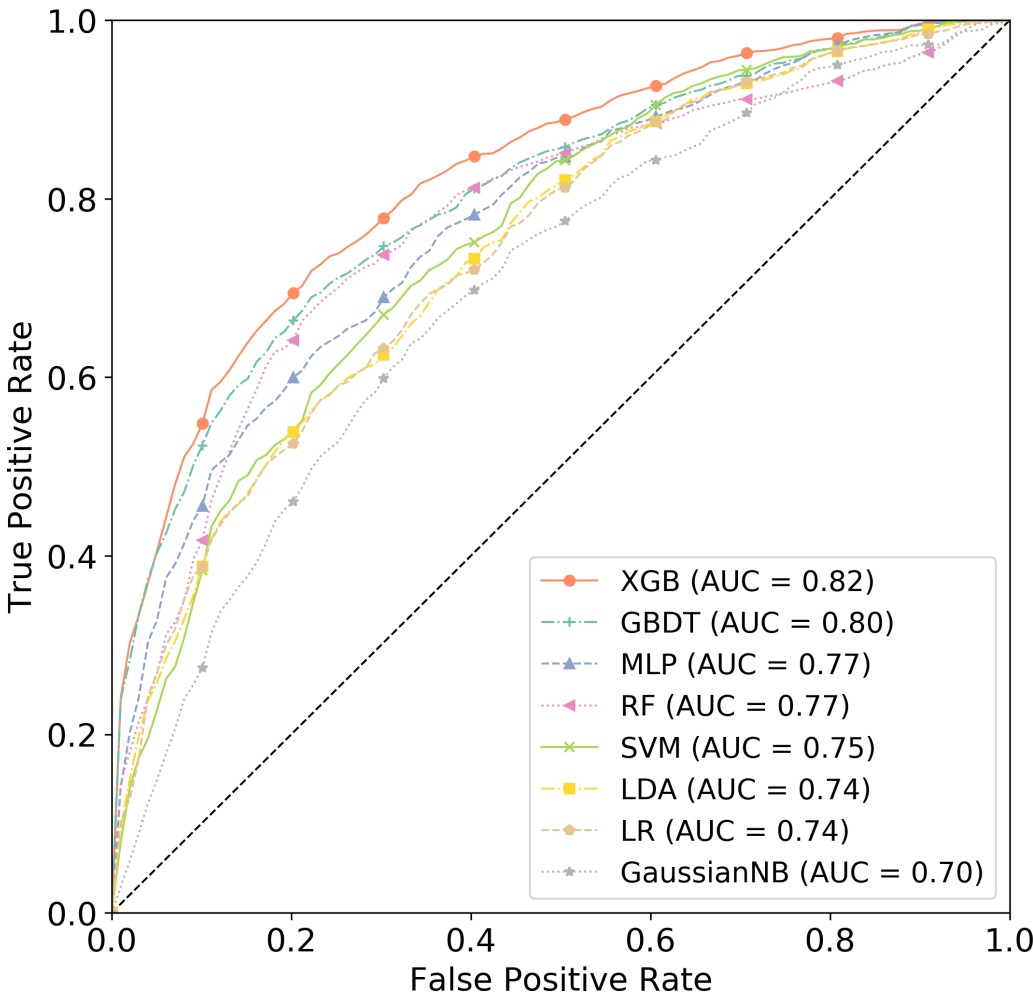

**Figure 3** ROC curves of several machine learning methods with parameters tuned on the training set to obtain an optimal model.

applied with three optimized parameters: learning_rate = 0.04, max_depth = 4, and colsample_bytree = 0.2.

To explore the possibility of further refining the features selected from mutual information, we examined the correlations among the 65 features. We identified several highly related features in UniProt, as highlighted in yellow in Fig. S1. Subsequently, we utilized the feature selection method with XGBoost (using default parameters) to determine the importance of the features. We employed sequential feature selection (SFS) and used the optimized parameters of XGBoost to train the data. Figure 4 illustrates the comparison of the AUC results for these features. The top 10 features (highlighted in bold in Table S2) achieved the highest mean AUC of 0.83 with 10-fold cross-validation. We conducted a performance comparison between the top 10 features and the absence of the top k (1 to 10) features using 10-fold cross-validation (Fig. 5). The results indicate that excluding features like ExonSnpDensity and ExonHapMapSnpDensity, which quantify

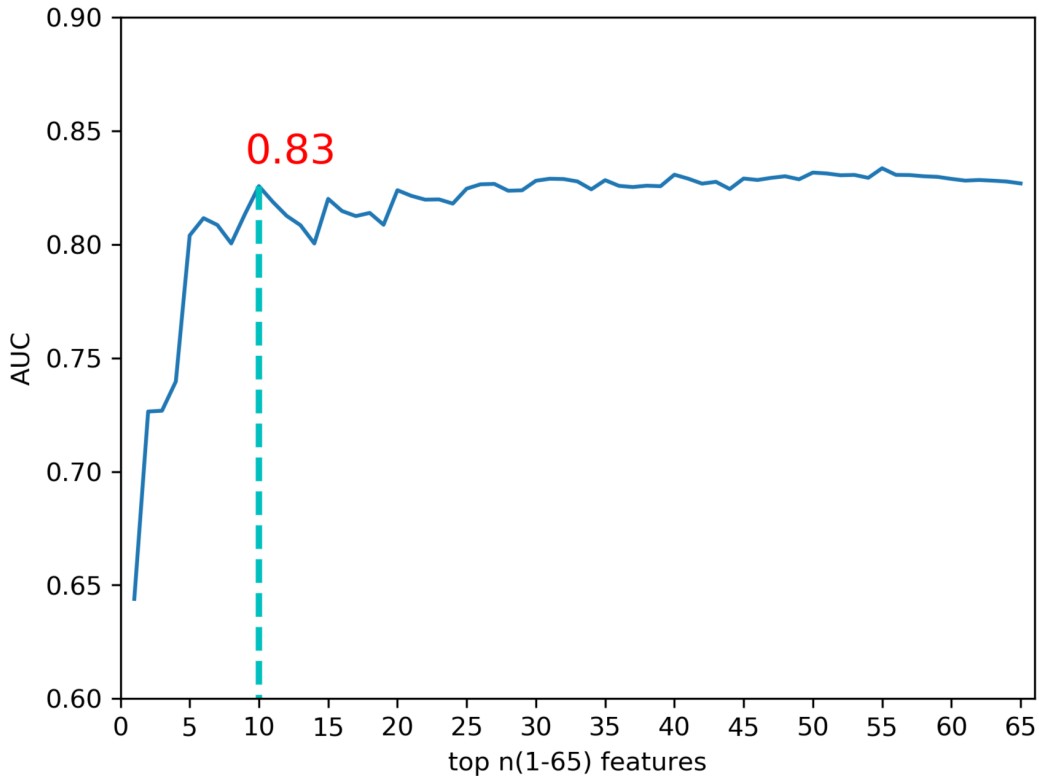

**Figure 4** **Comparison of AUC values with top n (1–65) features predicted by XGBoost feature importance on the training set.** The optimal model was achieved with the top 10 features on the training set, including 'UniprotDOM_PostModEnz', 'MGAPHC', 'UniprotCARBOHYD', 'UniprotREP', 'ExonSnpDensity', 'ExonConservation', 'UniprotMETAL', 'MGAEntropy', 'ExonHapMapSnpDensity', 'UniprotDOM_MMBRBD'.

the density of SNPs and HapMap-verified SNPs in exons, resulted in a notable 4.8% and 4.3% decline in prediction performance, respectively. Although classified as exon features in CRAVAT, these features also relate to evolutionary conservation—a factor significantly influencing cancer driver prediction performance (*Ostroverkhova, Przytycka & Panchenko, 2023*; *Nourbakhsh et al., 2024*; *Rogers, Gaunt & Campbell, 2021*).

Additionally, the feature UniprotMETAL, which relates to the binding of metal ions at mutation sites, is crucial given the role of metal ions as protein cofactors in cellular processes linked to cancer development (*Xu et al., 2016*; *Ge et al., 2022*). Lastly, UniprotREP, which denotes genomic repetitive regions, is highlighted for its potential to induce genomic instability—a hallmark of cancer genomes, thereby strongly correlating with cancer occurrence (*Criscione et al., 2014*; *Liao et al., 2023*). Consequently, we chose XGBoost with the top 10 features and optimal parameters as the final CDMPred model.

## Comparison with models trained on class labels using random permutation

To demonstrate that CDMPred acquired knowledge beyond random noise, we trained corresponding models of CDMPred_random. The mean values and standard deviations of

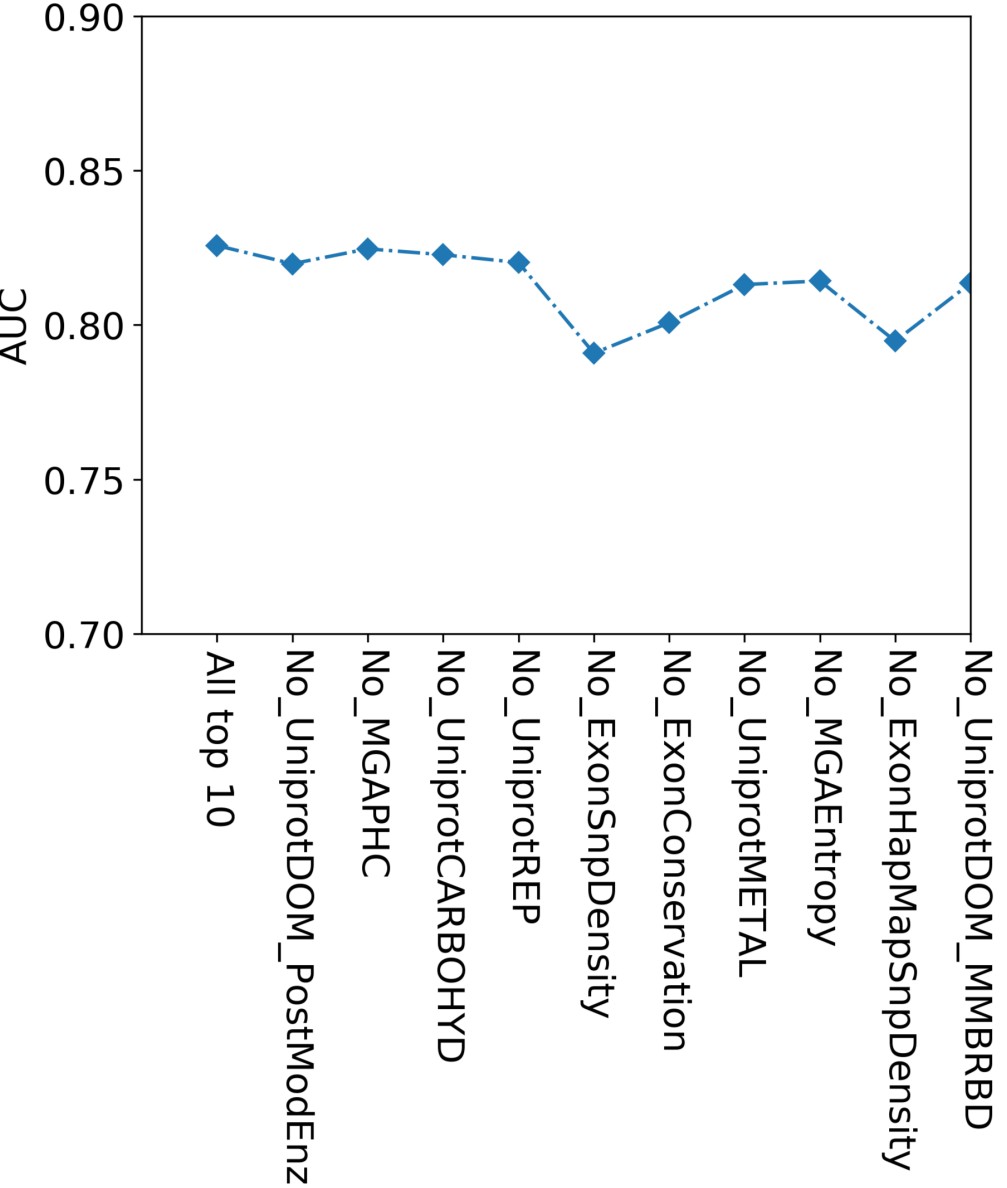

**Figure 5** Comparison of performance with the top 10 features and without the top k (one to 10) features by 10-fold cross-validation on the training set.

AUC values on the training set with 10-fold cross-validation are shown in Table S3. The results illustrate an AUC value of 0.826 for the original CDMPred model. Nevertheless, the AUC value experienced a significant decrease upon random permutation of class labels for training the CDMPred_random model. Additionally, CDMPred exhibited statistical solid significance (with a $p$-value <0.001) compared to other models. The computational setup involved a system with 16 GB of memory, an Intel(R) Core (TM) i7-9700 CPU operating at

3.00 GHz with eight cores and running on a 64-bit Windows 10 system. The permutation test incurred a time cost of approximately 966 s.

## Performance comparison with state-of-the-art predictors

To evaluate the performance of CDMPred on unseen samples, we assembled an independent test set. We utilized widely recognized tools designed explicitly for cancer-specific and general diseases, including CHASMplus, CHASM, CanDrA, FATHMM, TransFIC, and CScape-somatic. Additionally, we collected ten general disease predictors: SIFT (*Kumar, Henikoff & Ng, 2009*), Mutation Assessor (*Reva, Antipin & Sander, 2011*), PolyPhen-2 (*Adzhubei et al., 2010*), CADD (*Kircher et al., 2014*), MetaLR (*Dong et al., 2015*), MetaSVM (*Dong et al., 2015*), DANN (*Quang, Chen & Xie, 2015*), REVEL (*Ioannidis et al., 2016*), M-CAP (*Jagadeesh et al., 2016*), and MVP (*Qi et al., 2021*). For the cancer-specific methods, we submitted the test data to the respective websites of each tool to obtain the prediction results. As for the general disease predictors, we downloaded the dbNSFP4.1a software (https://sites.google.com/site/jpopgen/dbNSFP) and utilized a script written in Java to retrieve the prediction results from the database (*Liu et al., 2020*). All comparisons were conducted while disregarding any missing values from the tools. Figures 6 and 7 depict the ROC and PR curves, respectively. The results demonstrated that CDMPred exhibited the highest performance in terms of AUC and AUPR. The Delong tests (*De Long, De Long & Clarke-Pearson, 1988*) were conducted to assess whether the CDMPred's performance was significantly different from that of other cancer-specific methods (Table S4) and general-purpose methods (Table S5). The *p*-value of the AUC results indicated that CDMPred exhibited significantly superior performance to all cancer-specific methods and was superior to nine out of ten general-purpose methods, except CADD (*p*-value =0.09677, Delong's test). Furthermore, CDMPred demonstrated strong significance (with a *p*-value <0.001) compared to the other methods. It is worth noting that the AUPR value of CADD is 0.68 while that of CDMPred is 0.80. In total, the performance of CDMPred was robust.

## Case study

The principal advantage of our computational approach lies in its ability to significantly broaden the scope of analysis while concurrently preserving efficiency in terms of time and cost. A particularly compelling feature is its potential to inform and direct future experimental research, adeptly pinpointing candidate cancer driver mutations that merit in-depth investigation. In this context, we presented two illustrative cases predicted by CDMPred, juxtaposed with the predictions from several leading-edge methods. These include the cancer driver predictors CHASMplus and CScape-somatic, and the pathogenic missense mutation predictors ESM1b and AlphaMissense.

The kinase insert domain receptor (KDR), a type III receptor tyrosine kinase, is pivotal in mediating proliferation, survival, and migration induced by vascular endothelial growth factor. Its involvement is implicated in several diseases, including lymphoma (*Rotunno et al., 2016*). Experimental evidence has shown that p.A1065T, located within the activation loop, induces constitutive autophosphorylation on tyrosine independent of vascular
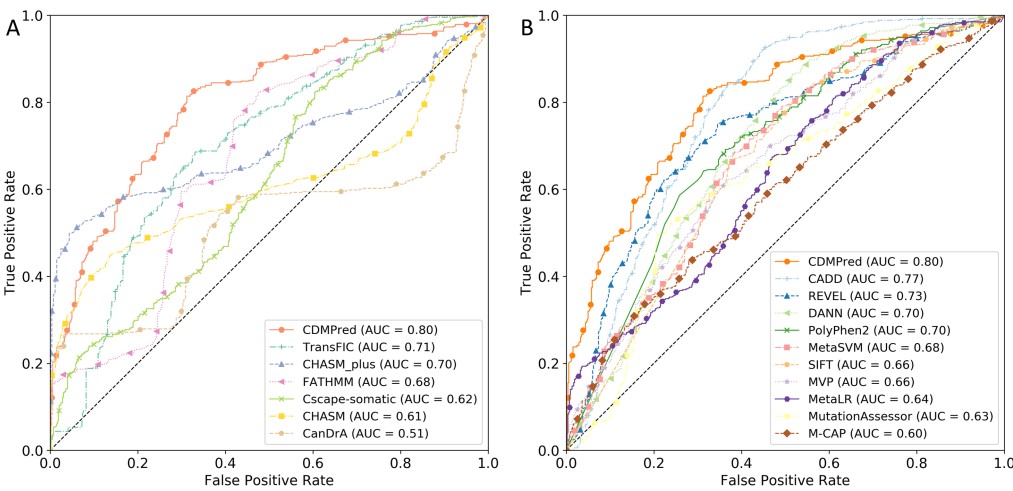

**Figure 6** **ROC curves of CDMPred relative to state-of-the-art models on the independent test set.** (A) Comparison of performance between CDMPred and other methods designed to predict single nucleotide driver variants in cancer. (B) Comparison of performance between CDMPred and several general-purpose predictors.

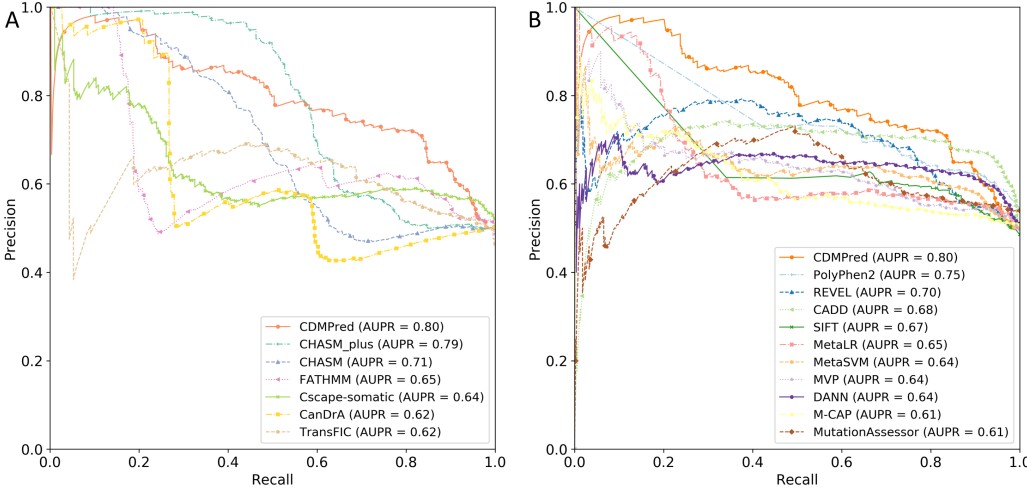

**Figure 7** **Precision–recall (PR) curves of CDMPred relative to state-of-the-art models on the independent test set.** (A) Comparison of performance between CDMPred and methods designed to predict single nucleotide driver variants in cancer. (B) Comparison of performance between CDMPred and several general-purpose predictors.

endothelial growth factor stimulation. Additionally, kinase inhibitors effectively suppressed its activity (*Rotunno et al., 2016*; *Flerlage et al., 2023*). Our computational approach, CDMPred, precisely identified the KDR-p.A1065T mutation as a significant driver with a high prediction score of 0.824. In stark contrast, the cancer driver predictors CHASMplus and CScape-somatic misclassified it as a passenger mutation, with substantially lower prediction scores of 0.119 and 0.139, respectively. The pathogenic missense mutation

predictors ESM1b and AlphaMissense also provided divergent assessments, with ESM1b categorizing it as a tolerated mutation (score = 0.423) and AlphaMissense as a likely benign mutation (score = 0.335).

The Mitogen-Activated Protein Kinase Kinase 1 (MAP2K1) gene encodes MEK1, a pivotal protein kinase in the RAS/MAPK pathway that transduces extracellular chemical signals to the cell nucleus. This signaling pathway regulates fundamental cellular processes such as proliferation, differentiation, migration, and apoptosis. A recent clinical observation identified the p.E120D mutation in a non-small-cell lung cancer patient (*Wang et al., 2021*). CDMPred and CScape-somatic correctly predicted MAP2K1-p.E120D as a significant driver mutation, with prediction scores of 0.810 and 0.742, respectively. Conversely, CHASMplus misclassified this mutation with a borderline score of 0.499, suggesting it was a passenger mutation. Additionally, ESM1b and AlphaMissense provided divergent classifications, with ESM1b scoring it as a tolerated mutation (score = 0.334) and AlphaMissense deeming it an ambiguous mutation (score = 0.366).

## DISCUSSION

For cancer-specific methods, TransFIC applied to PolyPhen-2 predictions due to the fewest missing values and achieved the second-highest AUC performance but ranked last in terms of AUPR. The CHASM prediction yielded an AUC of 0.61, sensitivity of 0.74, and specificity of 0.15. Similarly, CanDrA achieved an AUC of 0.51, sensitivity of 0.76, and specificity of 0.07. Therefore, both CHASM and CanDrA exhibited poor performance on the negative samples, indicating a severe imbalance that resulted in significantly low AUC values (as discussed below).

The CHASM training set comprised a balanced collection of positive and negative samples; however, there was only a 0.6% overlap at the transcript level (*Carter et al., 2009*). Therefore, we hypothesized that CHASM might be influenced by type 2 circularity, where the variant status was predominantly predicted based on other variants within the same protein (*Grimm et al., 2015*). As anticipated, 53% of false negatives in the CHASM predictions occurred in transcripts that completely overlapped with positive data in the CHASM training set. In contrast, only 0.9% were found in transcripts that entirely overlapped with negative data in the CHASM training set. Moreover, the opposite was observed for the true negatives of the CHASM predictions, with a higher number of samples found in transcripts that exclusively overlapped with negative data in the CHASM training set. Consequently, CHASM was influenced by type 2 circularity.

CanDrA proposed that driver mutations recurrently occurred in proximity (hotspots) in various types of cancer, whereas passenger mutations were not detected in any Cancer Gene Census (CGC) genes (*Mao et al., 2013*; *Futreal et al., 2004*). Based on our findings, we suspected the presence of type 2 circularity in CanDrA since it adhered to the screening criteria of the training set, resulting in minimal overlap between positive and negative samples at the transcript level. When the genes of the negative sample in the independent test set overlapped with the CGC genes, we identified shared genes in both sets. These genes were absent in the CanDrA training set, and the independent test set consisted of 95%

negative samples, of which only 3% were true negatives. Moreover, the genes exclusively present in the negative samples of the independent test set, which could potentially be the genes corresponding to negative samples in the CanDrA training set, comprised 5% of the negative samples in the independent test set, of which >80% were predicted to be true negatives. Therefore, CanDrA predicted the variant status by relying on other variants within the same protein, indicating the presence of type 2 circularity. We have shown that the low AUC values obtained by both CHASM and CanDrA can be primarily attributed to type 2 circularity. Furthermore, considering the quality of training data, we propose that negative samples used in CHASM and CanDrA fail to represent the broad spectrum of passenger mutations.

CDMPred demonstrated the highest comprehensive predictive capacity among the general-disease deleterious mutation predictors, followed by CADD, Polyphen-2, and REVEL. Interestingly, these methods also surpassed the second-best predictor specific to cancer. PolyPhen-2 achieved an AUPR of 0.75 and a sensitivity of 0.83, indicating a relatively higher predictive ability than CDMPred for positive samples. However, in both the positive and negative samples of the independent test set, numerous predictions made by PolyPhen-2 were classified as "positive", potentially corresponding to a range of diseases rather than solely cancer drivers (*Bertrand et al., 2018*). For instance, one of the true positives predicted by PolyPhen-2, "GATA2:p.R398W", is associated with acute myeloid leukemia and alveolar proteinosis (*Kazenwadel et al., 2012*; *Griese et al., 2015*).

Furthermore, one of the false negatives predicted by PolyPhen-2, "HMBS:p.D359N", is associated not only with cancer but also with acute intermittent porphyria (*Dorschner et al., 2013*; *Lewis, 2006*). Therefore, we directed our attention to the genes corresponding to the true negative and positive categories and the false negative and positive categories in the PolyPhen-2 predictions. We conducted enrichment analysis using the online tool DAVID to validate the suppositions mentioned above (*Huang, Sherman & Lempicki, 2009*). We gathered the pathways exclusively associated with general diseases, excluding cancer, and subsequently calculated the adjusted $p$-value ($<0.05$) using the hypergeometric test followed by the Benjamini–Hochberg test. Upon mapping the enrichment results at the mutation level, 65% of the results were associated with diseases present in both the true negatives and true positives of the PolyPhen-2 predictions. In comparison, 54% were associated with diseases present in both the false negatives and false positives of the PolyPhen-2 predictions. In conclusion, these findings support the presence of a systematic bias in driver mutation prediction by PolyPhen-2, even among general disease predictors.

CDMPred utilizes high-quality passenger mutations from dbCPM to distinguish between cancer missense driver mutations and passenger mutations. The results demonstrate that CDMPred achieved superior performance compared to various state-of-the-art methods for cancer-specific and general diseases. While our method offers significant insights, it has limitations. First, the curated datasets exhibit inherent biases, acknowledging that a mutation's role as a driver or passenger mutation can vary with tumor microenvironments, as noted in recent literature (*Ostroverkhova, Przytycka & Panchenko, 2023*; *Wodarz, Newell & Komarova, 2018*). Therefore, this introduces selection and information bias in our supervised learning model. Second, our current method lacks the exploration of advanced

machine-learning techniques. Recent studies have demonstrated that deep learning and protein language models could enhance performance in identifying pathogenic missense mutations (*Cheng et al., 2023*; *Schubach et al., 2024*).

## CONCLUSIONS

The predictive performance of machine learning methods relies heavily on the quality of the training data. Consequently, including well-defined positive and negative samples of known instances is crucial. This study introduces CDMPred, a novel predictor that distinguishes cancer missense driver mutations from passenger mutations. Specifically, high-quality passenger mutations from dbCPM, chosen for their superior representativeness in modeling the diverse range of passenger mutations, were utilized as negative samples in the training set. The results demonstrated that incorporating high-quality passenger mutations through an ensemble learning method enhanced the accuracy of algorithms in predicting driver mutations in human cancer. In the future, our research will expand to include a broader collection of experimentally verified negative samples and explore the utilization of ensemble deep learning methods further to refine the predictive model (*Xi et al., 2023*; *Deng et al., 2020*).

## ACKNOWLEDGEMENTS

The authors thank the members of our laboratory for their valuable contributions to CDMPred and the reviewers for their useful comments.

### Funding

This work was supported by the National Natural Science Foundation of China (U22A2038, 82101611, 32271283, and 62102004) and the Natural Science Foundation of the Anhui Higher Education Institutions of China (2023AH051392). The funders had no role in study design, data collection and analysis, decision to publish, or preparation of the manuscript.

### Grant Disclosures

The following grant information was disclosed by the authors:
the National Natural Science Foundation of China: U22A2038, 82101611, 32271283, 62102004.
The Natural Science Foundation of the Anhui Higher Education Institutions of China: 2023AH051392.

### Competing Interests

The authors declare there are no competing interests.

### Author Contributions

- Lihua Wang performed the experiments, analyzed the data, prepared figures and/or tables, authored or reviewed drafts of the article, and approved the final draft.

- Haiyang Sun performed the experiments, analyzed the data, prepared figures and/or tables, authored or reviewed drafts of the article, and approved the final draft.
- Zhenyu Yue analyzed the data, prepared figures and/or tables, and approved the final draft.
- Junfeng Xia conceived and designed the experiments, authored or reviewed drafts of the article, and approved the final draft.
- Xiaoyan Li conceived and designed the experiments, authored or reviewed drafts of the article, and approved the final draft.

## Data Availability

The raw data for training and testing and the code for CDMPred are available in the Supplementary Files.

## Supplemental Information

Supplemental information for this article can be found online at http://dx.doi.org/10.7717/peerj.17991#supplemental-information.

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
