# Peer review of "CDMPred: a tool for predicting cancer driver missense mutations with high-quality passenger mutations"

_PeerJ, doi:10.7717/peerj.17991_

## Round 0.1 · original submission · Minor Revisions

Please attend to the suggested improvements to the manuscript.

Reviewer 1 ·

Basic reporting

Li et al curated a balanced dataset and developed a cancer driver missense mutation predictor (CDMPred) employing feature selection through the ensemble learning technique XGBoost. The AUC values of CDMPred on the training and independent test sets are over 0.8, and the predictor showed superior performance compared to various state-ofthe-art methods.

Experimental design

The dataset and model construction are meaningful and could help dig out the features behind the principles of mutations.

Validity of the findings

The results are statistically reasonable, the manuscript is well written and the conclusion is consistent with their results.

Additional comments

1. Some previous important methods for the prediction of driver mutation are missing in their introduction (Am J Hum Genet. 2017, 100(1): 5-20; Nucleic Acids Res. 2019, 47: 315-321; Nucleic Acids Res. 2023, 51: 129-133)
2. Perspective prediction in the manuscript is absent, it is better to show or discuss the ability of this method in real cases.

·

Basic reporting

1. The paper should include more references on the most recent studies.

2. The font style is not consistent throughout the paper. It can be fixed for better readability.

Experimental design

1. It is better to provide more details on the feature engineering process and the rationale behind the selected features.

2. The data preprocessing description such as how to control confounding variables can be better elaborated.

Validity of the findings

1. The paper should discuss the potential limitations of the current study, such as the possibility of biases in the curated datasets, or the lack of exploration of machine learning techniques.

Reviewer 3 ·

Basic reporting

In this papers the authors present CDMPred, a new predictor of cancer-associated mutations, which unlike previous predictors was trained to differentiate cancer-associated from "passenger" (non-cancer-related) mutations, using the Cancer Passenger Mutation database (dbCPM) as a source of natural passenger mutations (they do discuss the fact that "passenger mutations" might not be a clearly-cut concept).

I found the paper well written, succinct, useful (the model compares favorably or equivalently to previous models) and interesting (in particular the identification of key mutation features that determine the predictor's calling). The Supplementary sections provide detailed descrip

My only remark is that the paper is focused on model building and comparison, and could have benefited from a discussion of the top 10 features identified during the predictor's training/analysis. Are these top 10 features surprising? Do they correspond to known cancer mechanisms?

Experimental design

The experimental seems sound to me. The authors tested many different ML models to ensure they built the best possible model (they do mention ensembling as a potential next step in the discussion), then they compare CDMpred to pre-existing methods, using well-established classification metrics (AUC).

Validity of the findings

The findings seem valid.

---

## Round 0.2 · Minor Revisions

Thanks for addressing all the technical issues pointed out by the reviewers. I propose several editorial suggestions in the attached PDF as tracked changes; please review them and consider including them in a final revised version.

---

## Round 0.3 · accepted · Accept

Thank you for addressing the editorial suggestions. Your manuscript has been accepted in PeerJ.